# Detection of Intraocular Hypertension during Opportunity Screening (Check-Up Medical Inspections)

**DOI:** 10.3390/jpm12050777

**Published:** 2022-05-11

**Authors:** Gabriel Zeno Munteanu, Zeno Virgiliu Ioan Munteanu, George Roiu, Cristian Marius Daina, Lucia Georgeta Daina, Mihaela Cristina Coroi, Carmen Domnariu, Daniela Carmen Neculoiu, Adrian Sebastian Cotovanu, Dana Badau

**Affiliations:** 1Faculty of Medicine and Pharmacy, University of Oradea, 410087 Oradea, Romania; gabi_munteanu_91@yahoo.com (G.Z.M.); zenomunteanu@yahoo.com (Z.V.I.M.); roiug70@yahoo.com (G.R.); cristi_daina@yahoo.co.uk (C.M.D.); lucidaina@gmail.com (L.G.D.); opticlar@gmail.com (M.C.C.); 2Faculty of Medicine, Lucian Blaga University, 550169 Sibiu, Romania; carmen.domnariu@ulbsibiu.ro; 3Faculty of Medicine, George Emil Palade University of Medicine, Pharmacy, Sciences and Technology, 540142 Targu Mures, Romania; dana_nec@yahoo.com; 4Faculty of Medicine, Transilvania University, 500068 Brasov, Romania; adrian.cotovanu2000@yahoo.com; 5Petru Maior Faculty of Sciences and Letters, George Emil Palade University of Medicine, Pharmacy, Sciences and Technology, 540142 Targu Mures, Romania; 6Interdisciplinary Doctoral School, Transilvania University, 500068 Brasov, Romania

**Keywords:** opportunity screening, check-up medical inspections, intra-ocular pressure, ocular hypertension, primary open angle glaucoma, secondary prevention, visual field

## Abstract

The aim of the study is the early detection of OHT (Ocular hypertension) in patients, in the activity of secondary prophylaxis (opportunity screening-medical check-up), to prevent blindness caused by POAG (Primary Open Angle Glaucoma). In Romania, medical examination of personnel with responsibilities in Transport Safety (TS) is legally regulated, being mandatory as a result of the internal transposition of European legislation in the field. The addressability of the patients for examination was determined by personal choice, depending on the accessibility of the medical service on the profile market (availability and price). The design of the study is epidemiological, observational, descriptive and retrospective. The standardized medical protocol included: personal medical history (anamnesis), physical ophthalmological examination, Intraocular pressure (IOP) measurement and Visual Field (VF) performance, with Automated Perimeter “Optopol PTS 910” through “Fast Threshold” strategy. The specialized medical research was completed with a dichotomous questionnaire entitled “Symptom Inventory”, made according to the recommendations of patients resulting from “Focus group” research. The study was carried out within the “Check-up” type medical controls upon request, only for personnel with positions in Transport Safety (TS), during January–December 2021 at S.C. ARTIMED S.R.L. Oradea, Bihor County. Health analysis was performed for 820 people, of whom 71 people (8.65%) tested positive for IOP > 21 mmHg, (suspected OHT) compared to 749 (91.35%) with normal values (Normal intraocular pressure-NIOP); the two lots being statistically significantly different (*x*^2^ = 560.590, *df* = 1, *p = 0.000*). The study involved 754 men (92.0%) and 66 women (8.0%), the sex ratio is 11.42 (Exp (B) = 0.782, Sig = 0.558, 95% CI = 0.343–1783; sex is not a significant predictor at the 5% level). The prevalence of OHT was 8.66% for the whole group, 8.48% for men and 10.60% for women. In the screening action for the whole group of patients the following was determined: IOP reference = 20.85 mmHg, Sensitivity (Se) = 91.5% and 1-Specificity (Sp) = 0.073, (Sp = 92.7%). The predictive value of the screening test was: Positive Predictive Value (PPV) = 90.1% and Negative Predictive Value (NPV) = 91.7%; Area under the ROC Curve (Receiver Operating Characteristic) = 0.986, Sig. = 0.000, CI_95%_ = 0.979–0.993. A binary logistical model of a questionnaire was developed to determine the screening parameters which significantly predicted OHT: IOP (OR = 4.154, 95% CI: 3.155–5.469), Age < 40 years (OR = 0.408, 95% CI: 0.239–0.698) and Pattern Defect (PD) (OR = 1.475% CI: 1.130–1.925). The results of this study assess health status through regular medical examinations, and highlight their importance and usefulness in secondary prevention activity. The particularity of this “check-up” type for personnel with attributions in transport safety is based on two essential aspects: the legal obligation to perform it and the fact it is financed by the beneficiary (the employer). In patients suspected of OHT after antiglaucoma treatment, IOP statistically significantly decreased.

## 1. Introduction

Intraocular hypertension is a condition in which the IOP is above the normal range (>21 mmHg without treatment), but without any evidence of structural or functional damage to the optic nerve. [1]. The risk of OHT increases with age, and vision does not change until it progresses to glaucoma. High IOP is a predictor of conversion from OHT to POAG, and the Relative Risk (RR) is estimated to be 1.1 for every 1 mmHg increase in initial IOP [1]. The prevalence of OHT increases with age, from 1.25% in the population between 30 and 39 years to reach 10.5% in the population between 70 and 79 years [2]. Glaucoma comprises a heterogeneous group of eye diseases of multifactorial etiology, manifested by specific structural and functional changes of the optic nerve, which can cause blindness. Prophylaxis for OHT and POAG includes the following general medical purposes: promotion, maintenance, and recovery of vision to the greatest extent possible [3,4].

Screening is defined as “a secondary prophylaxis action aimed at identifying presumed persons affected by a previously unknown latent health problem by performing a test, examination, or other investigative techniques that can be quickly applied on a mass scale” [5]. Early disease detection can be accomplished through the following medical procedures: passive–active detection during current medical consultations, regular medical examinations at nodal ages, examination of people at risk, and “Check-ups” or health checks on request (for reasons other than the disease assessed in specialized services at the beneficiary′s initiative and/or as a legal obligation) [5,6,7]. “Health check on request screening is as effective as mass screening, but it is limited to the population that attends health services” [8,9]. 

The screening is followed by two phases: the “diagnostic phase”, in which likely patients are subjected to a diagnostic test to confirm the disease, and the “therapeutic and medical surveillance phase”. Universal glaucoma screening of the general population is not cost-effective and current practice targets high-risk groups [5,10,11].

In Romania, since 2013, the Ministry of Transport and Infrastructure and the Ministry of Health have imposed the obligation of periodic medical examinations for positions with attributions in ST: training and professional development courses; employment, regular examination and change of function.

The aim of the study is the early detection of OHT in patients, in the activity of secondary prophylaxis to prevent blindness caused by POAG. The objectives of the study were to detect and identify particular aspects of patients′ health status through epidemiological, clinical, functional and statistical evaluation, as well as to evaluate the protocol and the results (effectiveness) of the activity.

## 2. Materials and Methods

### 2.1. Ethical and Legal Issues

The study was conducted after obtaining: “Ethical Opinion for Access in scientific interest to archived patient data” no. 108/31.12.2019, within S.C. ARTIMED S.R.L., Oradea, Bihor County, Medical Unit Approved by the Ministry of Transport and Infrastructure, Certificate approval no. 919/22.04.2019) [6,12]. S.C. ARTIMED S.R.L has been carrying out this type of medical activity since 2014 with a portfolio of 8452 patients investigated.

### 2.2. Data Collection

The database was set up strictly according to the documents (existing records) recorded in the “Personal Medical Record for Transport Safety” from the archive of S.C. ARTIMED S.R.L. Oradea (personal medical history data, objective examination and, for ophthalmology, the specialized technical investigations: IOP and VF). Additionally, the results of the ”Inventory” questionnaire, completed by the patients were recorded [12,13].

### 2.3. Study Design

The research was designed based on an epidemiological, observational, descriptive, retrospective study.

### 2.4. Methodology

The screening had a legally required periodicity of one year and consisted of examinations in seven medical specialties: Internal Medicine, Surgery, Otorhinolaryngology, Ophthalmology, Neurology, Psychiatry and Laboratory Medicine [6,7]. Informed consent to participate in the research was obtained in writing from the participants, who knew the purpose and content of the study. All patients who presented at the medical check-up were professionally active in the field of TS: road and rail transport of people and goods. The criteria for inclusion in OHT syndrome were: IOP > 21 mmHg, no history of glaucoma, no specialist treatment, VF with no characteristic changes of POAG, no history of eye disease, no history of steroid treatment and no ocular surgery.

Exclusion criteria covered other forms of open-angle glaucoma: primitive juvenile glaucoma, secondary glaucoma (pseudo-exfoliative, pigmented, with crystalline particles, associated with intraocular, uveitic, neovascular hemorrhage, associated with intraocular tumors, associated with retinal detachment, post-traumatic, iatrogenic induced by corticosteroids and surgical treatment and/or laser). Other eye diseases, such as corneal, lens, vitreous and retinal diseases, etc., were excluded.

All persons investigated during the medical examination presented a medical certificate issued by their family doctor, which clinically certified the optimal state of health, for functions in TS, and lack of a history of ocular pathology. The epidemiological, demographic and ophthalmological parameters were used to characterize the health conditions of the investigated patients. The epidemiological parameters were: number of cases (people with a positive screening result—suspected OHT), age, sex, sex ratio, place of residence, level of education and marital status. 

The examination of the patients was performed by two methods: the interview (personal medical history and questionnaire) and by specialized medical investigation. For the ophthalmology specialty, the standardized medical protocol included: medical history of eye conditions, and physical eye examination that investigated the following: visual acuity, binocular vision test by monocular suppression (Worth test), oculomotor balance test in primary position (Cover test), eye motility, funduscopic examination, chromatic sense (Isihara test), contrast sensitivity (Pelli-Robson tables) [14].

The ocular functional examination continued with the determination and recording of the IOP with a pneumatic non-contact tonometer, the “NC Reichert R7 noncontact tonometer”. IOP was considered an important indicator both for the detection of OHT and in monitoring its progression under treatment. The IOP determination was performed at two times: in the morning between 8–10 and then between 13–14. Each measurement expressed the average of three trials for each eye. The most favorable result was chosen. The determination of the visual field was done with the computerized perimeter, the “Optopol PTS 910-Automated Perimeter” with the strategy “Fast Threshold”, using optical correction as needed. The following were considered: credibility indices (false positive and false negative answers), calculated HOV level, the theoretical inclination of the visual slope to 10°, structural defect PD (localized defects), average defect AD (generalized defects with decreased retinal sensitivity) and defect graph (“Bebie Curve”) [15]. 

The “Optopol PTS 910-Automated Perimeter” calculates, statistically and graphically, the centralized defect of the test result. ”The Bebie Curve” (indicating VF quality) is constructed by ordering the sensitivities of each tested point in descending order. According to the instruction manual, the “Bebie Curve” is particularly useful in quickly and immediately classifying the type of defect; especially in differentiating a diffuse loss from a localized or mixed one. [16].

For the statistical interpretation of the graph of the centralized defect of a test result (“Bebie Curve”) we used the following categorical classification (Table 1) [15].

In the study, the cases were admitted according to credibility indices, establishing a percentage threshold with additional qualitative descriptions. At the end of the Check-up medical examinations, a “Medical Opinion” was issued, which, together with a “Psychological Opinion”, documented the issuance of the work aptitude certificate by the Labor Medicine specialist [17]. 

The medical research was completed with a sociological tool, a dichotomous questionnaire, “Symptom Inventory”, resulting from “Focus group” research. Patients suspected of having OHT were referred to specialist ophthalmologists for confirmation of diagnosis, monitoring, counseling and, depending on adherence, specific treatment was started (Figure 1).

### 2.5. Statistical Analysis

Descriptive statistics included the study of central trend indicators (average), spread indicators standard deviation (SD). Continuous variables are presented as mean ± SD or percentage values. The ordinal variables were analyzed in terms of the normality of the distribution by performing the Kolmogorov-Smirnov and Shapiro-Wilk tests and nonparametric tests: U Mann-Whitney Test, Kruskal-Wallis Test and Wilcoxon Test for related scores. Categorial variables were described as frequencies and the Chi-Square Test was used to compare percentages between groups and evaluate the existence of a significant difference between two or more samples.

To discover probabilities, binary logistic regression was used to create a model in which the values of the criterion (dependent variable) were probabilistically related with the values of the predictors (independent variables). Correlations were investigated, and the relationship between the variables was assessed according to the value of the correlation coefficient as: strong (0.8–1), moderate (0.5–0.8), weak (0.2–0.5) and negligible (0–0.2). The evaluation of the intervention of the event was done by testing the statistical significance. Nonparametric correlation analysis was applied, calculating correlation coefficient, Spearman′s rho.

Crude odds ratios (cORs, considering one independent and one dependent variable) were used to estimate associated characteristics of each independent variable from the screening. Additionally, 95% confidence intervals (CIs), *x*^2^ values, and *p* values were calculated for each cOR. The OR (odds ratio) value means that the risk of getting the disease is higher in those who are exposed than in those who are not exposed. A binary logistic model was also developed to determine which screening parameters significantly predicted OHT. For the analysis of the questionnaire with dichotomous answers frequency tables, the Mann-Whitney Test and logistic regression analysis were used. The *p* value < 0.05 was considered statistically significant.

Statistical analysis was performed with the program IBM SPSS Statistics Version 22 [18,19,20,21,22,23].

## 3. Results

The study was conducted between January and December 2021. The health analysis was performed for 820 people, of which 71 people (8.65%) tested positive for OHT (IOP > 21 mmHg) and 749 people (91.35%) with NIOP; the two groups being statistically significantly different (*x*^2^ = 560.590, *df* = 1, *p = 0.000*). Also, 114 people (13.9%) had VF changes (not specific for POAG) of which 71 (62.2%) people had IOP > 21 mmHg and 43 (37.8%) with NIOP. The classification of patients, according to the risk of IOP values, was: 765 people (92.20%) without risk with IOP < 21 mmHg; 14 people (1.71%) with low IOP risk = 22–23 mmHg, 43 people (5.24%) with moderate IOP risk = 24–29 mmHg; 7 people (0.85%) at high risk IOP > 30 mmHg [24]. The structure of the functions for TS designates 699 persons (85.25%) professional drivers, 41 persons (5.00%) driving instructors, 78 persons (9.51%) transport managers and 1 retired in activity as driving instructor (0.24%) (Table 2).

From the family medical history recorded in the monitoring sheets of patients with HTO, can be retained: hypertension (first degree relatives) with a frequency of 13 people 1.58% (11 men and 2 women) and diabetes mellitus (first degree relatives) with a frequency of 8 people: 0.97% (5 men and 3 women). Pathological personal history records: hypertension with a frequency of 8 people 0.97% (7 men and 1 women) and diabetes mellitus with a frequency of 7 people: 0.85% (5 men and 2 women). Ophthalmological pathological personal history records: myopia in 9 people 1.09% (7 men and 2 women), hypermetropy in 6 people 0.73% (6 men) and “painful red eye” in 2 people: 0.24% (2 men). Repeated screening was chosen for patients suspected of having OHT.

To classify patients as individuals in one of the study groups, the arithmetic mean of the IOP between the right eye (Re) and the left eye (Le) was calculated, and the IOP value > 21 mmHg was considered the threshold for a positive screening result (suspicion of OHT) [24]. Starting from these premises, the prevalence of OHT in the studied group was calculated, this being 8.65% (71 cases). The positivity threshold of the test was chosen according to the context, at a value close to the definition of OHT (IOP > 21 mmHg), which consequently determines the start of treatment [24,25]. 

For the whole group (Re + Le) the reference IOP = 20.85 mmHg was determined, Se = 91.5% and 1-Sp = 0.073, (Sp = 92.7%). The predictive value of the screening test was: PPV = 90.1% and NPV = 91.7%. For the whole group of patients “Area under the ROC Curve” (ROC Curve–Receiver Operating Characteristic) = 0.986, Sig. = 0.000, CI_95%_ = 0.979–0.993.

The evaluation of the “Check-up” type medical controls upon request showed that 195 persons (23.78%) were examined for employment, 181 persons (22.07%) for schooling, 444 persons (54.15%) for regular medical control and no person for change of position for TS. 

All patients with a positive screening result, suspected of having OHT, received ocular hypotensive treatment with antiglaucoma drugs (prostaglandin analogues) in combination with neuroprotective drugs, without undergoing surgery. After treatment, IOP decreased statistically significantly: by 4.56 mmHg (17.92%); from 25.44 ± 3.04 mmHg to 20.88 ± 3.15 mmHg (*p* = 0.000); (Table 3). 

The ocular functional examination included the determination of visual acuity (VA) and analysis of VF parameters. At the time of examination of patients suspected of having OHT, 29.58% required optical correction. The interpretation of the results at the computerized perimeter was done by analyzing the credibility indices and the VF parameters (Table 4.). 

The statistical study of the differences between VF parameters in patients with suspected OHT and in patients with NIOP are presented in Table 5. The values recorded for “calculated HOV” are statistically significant in both groups (*p* = 0.014) and also for “AD” (*p* = 0.001).

In assessing the visual function, the “Babie Curve” graph was considered for the rapid assessment of the integrity of the visual field in relation to age (Table 6).

The distribution of the centralized defect resulting from the VF (Bebie Curve) examination shows a predominance of type III and type V models for both OHT and NIOP suspects. The Chi-Square Test, showed a statistically significant difference between the types of “Bebie Curve” indicators for Re and Le: in OHT: Re: *x*^2^ = 156,394, *df* = 4, *p =* 0.000; Le: *x*^2^ = 128,366, *df* = 4, *p =* 0.000), and those with NIOP: Re: *x*^2^ = 1,637,295, *df* = 4, *p =* 0.000; Le: *x*^2^ = 1,597,963, *df* = 3, *p =* 0.000). The analysis of the correlation coefficients between the variables of VF parameters in patients with a positive test result (suspected for OHT) and in those with NIOP are presented in Table 7.

Also, using a questionnaire, we developed a binary logistic model to determine which screening parameters predicted significant OHT (Table 8).

The final regression model states that IOP is the most important risk factor, being a significant predictor at the level of 5%, (Exp.(B) = 4154, 95% CI = 3155–5469), along with “age over 40 years” (Exp.(B) = 0.408, 95% CI = 0.239–0.698). Also, the analysis of the regression model of VF parameters specifies a probabilistic association of OHT with predictor values at a statistically significant level: PD: Exp.(B) = 1.475, *Sig = 0.004,* 95% CI = 1.130–1.925; Slope at 10^0^: Exp.(B) = 1.397, *Sig = 0.003,* 95% CI = 1.121–1.742; False negative: Exp.(B) =1.032, *Sig = 0.003,* 95% CI =1.011–1.053 and AD Exp.(B) = 0.332, *Sig = 0.006,* 95% CI =0.152–0.726. Questionnaire Assessment, “Symptom Inventory”, shows the comparative distribution of affirmative responses (certifying the presence of the symptom) within the OHT and NIOP groups (Table 9).

In all cases, the Mann-Whitney U Test results were statistically significant (*p* = 0.000). “All the symptoms investigated were present in a higher absolute value in the group of OHT suspects, except for “Difficult to see at a distance.” The average age of the participants was 43.27± 9.92, which is consistent with the age at which refractive issues necessitate some form of optical correction.” 

We point out that the minimum essential distance visual acuity standards, both with and without optical correction, are stringent in order to gain ophthalmology approval [12]. Despite the fact that not all patients mentioned difficulties with distance vision in the questionnaire, statistical analysis of the VA test results shows that 235 patients (28.5%) of those tested required optical correction, with 25 patients (10.6%) having OHT compared to 210 patients (89.4%) without OHT.

Tearing is the dominant non-visual symptom, present in 34 people (4.15%) suspected of having OHT compared to 15 people (1.83%) with NIOP, with a gap of 2.23%; and dry eye has a gap of 0.85%. For the visual symptoms in OHT patients, the positive response to the symptom “Sensation of intraocular pressure” achieved the largest difference between the two groups: 1.59% (15 people, 1.83% in OHT suspects compared to 2 people 0.24%). The variables in the symptom questionnaire were considered predictors for a statistical model that specifies the individual characteristics associated with morbid conditions. Binomial logistic regression analysis with the “Enter” method is illustrated in Table 10. 

The accuracy of the classification was 45.1% for OHT suspects and 98.8% for those with NIOP, with an overall accuracy of 94.1%. Given the values of Exp.(B), the final regression model states that: “Eye strain” and “Tearing” are significant predictors at 5%. The probability of a person falling into the suspicious OHT category increases for “Eye Tension” by 8.5% and for “Tearing” by 7.0%.

## 4. Discussion

In Romania, models of morbidity and mortality show an increase in the prevalence of chronic diseases and mortality, due to biological, environmental, and behavioral risk factors and socio-economic conditions and health services [26]; particularly in the context of the elderly population. Medical examination for functions with attributions in TS is obligatory [6,7,12].

For ophthalmology, OHT is an important issue. Once it is discovered, it needs to be monitored. The main goal of OHT management is to prevent progression to POAG. Risk assessment, and structural and functional testing, guide disease management in an efficient, safe and cost-effective way [27]. The term OHT has been used for more than 30 years. Increased intraocular IOP is the main risk factor for conversion to POAG [28].

In preventive activities for early detection of OHT and POAG patients should be instructed to report for routine check-ups even in the absence of symptoms. They need to know essential information about the disease, and its severe and irreversible consequences. For safe and effective diagnosis and treatment, it is essential to perform examinations and monitor patients, due to the existence of functional, structural changes, or both [29]. For many adults with eye disease, organizing simultaneous eye exams with specific health checks can be a beneficial and effective measure to prevent vision impairment [30]. OHT screening strategy must be useful, tailored to patient requests, and acceptable [31]. 

Its early detection in primary care should be carried out at least in at-risk groups [32]. Usefulness of a glaucoma and ocular hypertension screening strategy in primary care OHT screening programs should primarily ensure vision preservation. There must also be an infrastructure that allows positive tested people to have access to specialist investigation and treatment services. Careful assessment of the costs and benefits of screening is required before widespread adoption of any program [33]. The population must be encouraged to take part in the examinations, and the programs must develop innovative strategies and approaches [34]. Every year, in March, worldwide, for a week, glaucoma is brought to the forefront through scientific awareness of the danger of irreversible vision loss [35]. The evaluation of the results of the screening activity, of short-term efficiency, calls into question the achievement of this activity in correlation with the number of detected and treated cases, the acceptability and the necessary costs [4].

Visual field testing, supplemented with new techniques, provides information that can detect disease progression and improves clinical decision-making, being useful for monitoring patients and reducing the burden of disease [36]. Elevated IOP values in patients with suspected OHT justify hypotensive treatment [37]. Delaying treatment in OHT can lead to a significant increase in the incidence of POAG, noting that the cumulative proportion of POAG development after 13 years is 0.22 in the untreated cohort (with a 7.5-year treatment delay) compared to 0.16 in the treated cohort [38]. IOP is the only modifiable risk factor in patients with POAG, and its decrease prevents the progression of the disease, emphasizing the importance of early diagnosis and treatment, and is an effective intervention [39,40,41,42]. Early treatment with an individualized approach requires active monitoring and rigorous medical therapy for all medical diseases [43,44,45] and especially in accordance with our study for IOP. The treatment of glaucoma is a continuous post-diagnosis process throughout life [1]. 

Limitations: for the screening procedure described in this paper, the examination protocol was strictly observed only in the normative act, subsequently depending on the case; the medical research included other specialized investigations, in each case, to establish the final diagnosis and treatment for staff in charge of transport safety [12,14].

The current screening strategy has three key advantages: the obligation to conduct regular examinations, beneficiary funding, and a complex and unitary examination protocol for all medical units (in Romania, the Ministry of Transport and Infrastructure approved a total of 1210 medical and psychological units from 2014 to 2022) [46]. The specification of the diagnosis of OHT/POAG is no longer limited by mandatory procedures. 

The novelty that this type of screening could bring in medical literature would be provided by the opportunity of a specialized study at national level that would benefit from four great advantages: unitary methodology, obligatory application at national level, comparability of statistical data, the unitary evaluation of the health condition of the personnel in this field of activity. 

State intervention was exercised only at three levels: the issuance of the normative act, the approval activity of the medical units and the supervision of their activities through the annual approval procedure.

## 5. Conclusions

We consider that the results of this study show that assessing state of health, through appropriate medical examinations, is useful in the activity of secondary prevention. The particularity of this “check-up” type control of personnel for functions with responsibilities in transport safety is based on two essential aspects: the legal obligation to perform it and financing from the beneficiary (the employer). In patients suspected of OHT after antiglaucoma treatment, IOP statistically significantly decreased. Screening for OHT should be uniformly integrated into curative–prophylactic medical care.

## Figures and Tables

**Figure 1 jpm-12-00777-f001:**
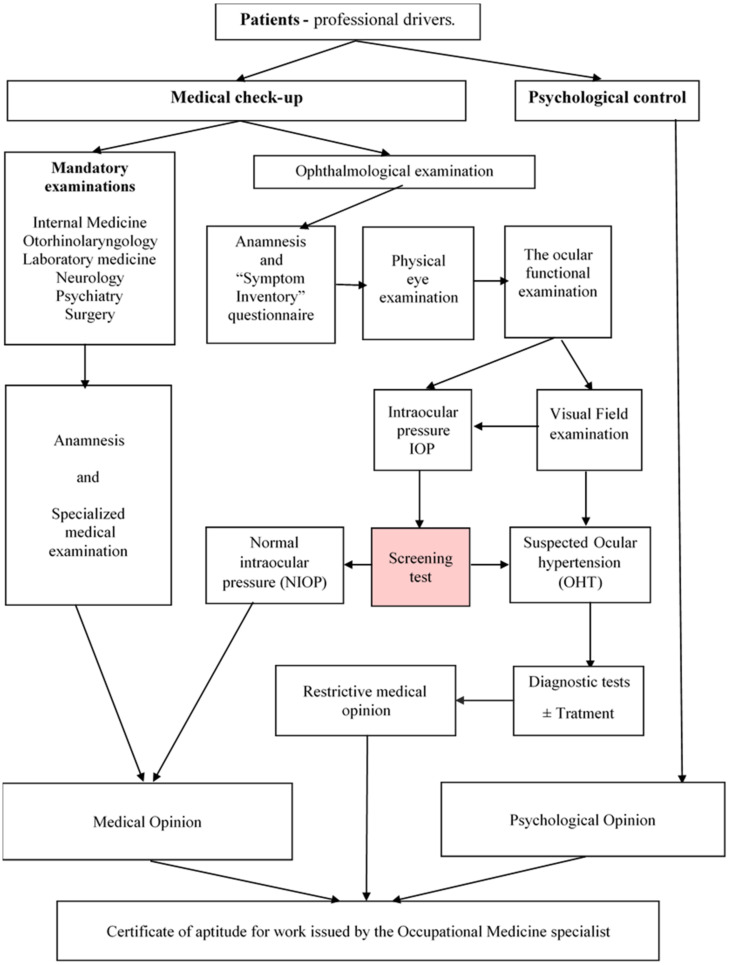
Staging research on the detection of OHT Screening (Red box—Screening test, highlights the main point of the screening action).

**Table 1 jpm-12-00777-t001:** Classification of the centralized defect of a “Visual Field” (VF) test result (Bebie Curve).

Bebie Curve Type I	Extensive and Deep Damage to the “Visual Field”
Bebie Curve type II	No real defects in the “Field of View”–happy trigger
Bebie Curve type III	Small but deep defects of “Visual Field”
Bebie Curve type IV	A “Visual Field” with a very large and shallow defect
Bebie Curve type V	A “Visual Field-Almost Normal”

**Table 2 jpm-12-00777-t002:** Distribution of demographic indicators of patients.

Parameters		Results					
		Patients Suspected of Having OHT		Patients with Normal IOP		Total Patients	
		*n*	%	*n*	%	*n*	%
**Number of Cases**		71		749		820	100
Sex	Male	64	90.14	690	92.12	754	91.95
	Female	7	9.86	59	7.88	66	8.05
Age		43.27 ±9.92	min = 18 max = 62	39.60 ±10.90	min = 18 max = 64	39.92 ±10.86	min = 18 max = 64
Residence	Urban area	50	70.42	623	83.18	673	82.07
	Rural area	21	29.58	126	16.82	147	17.93
Marital status	Married	43	60.56	513	68.49	556	67.80
	Un-married	21	29.58	153	20.43	174	21.22
	Widow	1	1.41	34	4.54	35	4.27
	Divorced	6	8.45	49	6.54	55	6.71
Studies	Primary Cycle	4	5.63	28	3.74	32	3.90
	Gymnasium cycle	3	4.23	52	6.94	55	6.71
	Professional school	10	14.08	132	17.62	142	17.32
	High school	26	36.62	307	40.99	333	40.61
	Post High school	3	4.23	111	14.82	114	13.90
	Higher education	20	28.17	76	10.15	96	11.71
	Post -university	5	7.04	43	5.74	48	5.85
Activity status	Professional driver	65	91.55	634	84.65	699	85.25
	Driving instructor	5	7.04	36	4.81	41	5.00
	Transport manager	0	0.00	78	10.41	78	9.51
	Retired car instructor	1	1.41	1	0.13	2	0.24

*n*—number, %—percent, min—minimum, max—maximum.

**Table 3 jpm-12-00777-t003:** Distribution of IOP parameters in OHT suspects patients, at the first and last consultation.

Parameters	Initial Consultation	Final Consultation	z	*p* *
IOP-Re	25.36 ± 3.21	21.46 ± 3.78	−3.432 ^b^	0.001
IOP-Le	25.52 ± 3.81	20.25 ± 3.29	−3.622 ^b^	0.000
IOP–Re + Le	25.44 ± 3.04	20.88 ± 3.15	−3.621 ^b^	0.000

^b^. Based on positive ranks, * Wilcoxon Test.

**Table 4 jpm-12-00777-t004:** Distribution of credibility indices in the interpretation of the “Visual Field” result.

Credibility Indices	Patients Suspected of Having OHT	Patients with IOP within Normal Limits (NIOP)	z	*p **
Average duration (minutes) Re	6.64 ± 1.49	7.12 ± 8.01	−0.906	0.365
Average duration (minutes) Le	6.47 ± 1.15	6.75 ± 1.42	−1.315	0.189
Average duration (minutes) Re + Le	6.56 ± 1.18	6.93 ± 4.10	−1.554	0.120
False positive-Re	3.59 ± 7.39	4.15 ± 6.53	−1.125	0.260
False positive-Le	4.52 ± 8.55	4.00 ± 6.12	−0.163	0.871
False positive-Re + Le	4.05 ± 6.92	4.07 ± 4.82	−0.952	0.341
False negative-Re	7.23 ± 10.25	4.93 ± 7.01	−1.348	0.178
False negative-Le	6.03 ± 8.17	4.49 ± 6.96	−1.395	0.163
False negative-Re + Le	6.62 ± 7.21	4.70 ± 5.25	−1.670	0.095

* Mann-Whitney U Test.

**Table 5 jpm-12-00777-t005:** Distribution of “visual field” parameters in OHT suspects and NIOP patients.

Parameters	Patients Suspected of Having OHT	Patients with IOP within Normal Limits (NIOP)	z	*p **
Tested points-Re + Le	295.01 ± 40.56	295.09 ± 40.54	−0.193	0.847
Visual slope at 10°-Re + Le	2.91 ± 0.71	2.87 ± 0.82	−0.367	0.714
Calculated HOV-Re + Le	24.16 ± 2.47	24.93 ± 2.34	−2.460	0.014
PD (Pattern defect)-Re + Le	0.46 ± 0.60	0.34 ± 0.38	−1396	0.163
AD (Average defect)-Re + Le	−0.19 ± 0.9	−0.02 ± 0.18	−3.221	0.001

*** Mann-Whitney U Test.

**Table 6 jpm-12-00777-t006:** Distribution of the types of “Bebie Curve” from the “Visual field” examination for suspected OHT and NIOP patients.

Bebie Curve Modes	Suspected OHT Patients				NIOP Patients			
	Re		Le		Re		Le	
	*n*	%	*n*	%	*n*	%	*n*	%
Bebie Curve type I	2	2.8	3	4.2	0	0.0	3	0.4
Bebie Curve type II	2	2.8	3	4.2	1	0.1	4	0.5
Bebie Curve type III	56	78.9	52	73.2	576	76.9	570	76.1
Bebie Curve type IV	2	2.8	3	4.2	9	1.2	9	1.2
Bebie Curve type V	9	12.7	10	14.2	163	21.8	163	21.8
Total	71	100	71	100	749	100	749	100

*n*—number of cases, %—percent.

**Table 7 jpm-12-00777-t007:** The comparative distribution of positive and negative correlation coefficients between VF parameters in OHT suspects and NIOP patients.

Correlations	Corelated Variables	Suspected OHT Patients Correlations		NIOP Patients Correlations	
		Spearman′s rho	Sig. 2-tailed	Spearman′s rho	Sig. 2-tailed
Pozitive correlation coefficients between VF parameters in OHT suspects and NIOP pacients	Average test duration-Average of the tested points	0.824 **	0.000	0.782 **	0.000
	Average Slope at 10°-False Negative Average	0.446 **	0.000	-	-
	Average test duration-PD Average	0.429 **	0.000	0.261 **	0.000
	AD Average-False Positive Average	0.387 **	0.001	0.094 **	0.010
	HOV Calculated Average-AD Average	0.376 **	0.001	0.466 **	0.000
	Average of the tested points-PD Average	0.367 **	0.002	0.345 **	0.000
Negative correlation coefficients between VF parameters in NIOP patients	Average test duration-AD Average	-	-	−0.186 **	0.000
	Average of the tested points-AD Average	-	-	−0.154 **	0.000
	PD Average-AD Average	-	-	−0.125 **	0.001

** Correlation is significant at the 0.01 level (2-tailed).

**Table 8 jpm-12-00777-t008:** Distribution of binomial logistic analysis results for OHT suspect variables at the screening test.

Risk Factor	Parameter Estimate	SE	Wald ꭓ^2^	*df*	*Sig*	Exp(B)	95% CI	
							>Lower	>Upper
IOP (1640 eyes)	1.424	0.140	102.931	1	0.000	4.154	3.155	5.469
Sex (male/female)	−0.246	0.421	0.343	1	0.558	0.782	0.343	1.783
Age (>40 years/<40 years)	−0.896	0.274	10.712	1	0.001	0.408	0.239	0.698
Hereditary history of diabetes	−1.882	0.741	6.445	1	0.011	0.152	0.036	0.651
Hereditary history of hypertension	−3.297	0.615	28.716	1	0.000	0.037	0.011	0.124
History of diabetes	−2.698	0.774	12.138	1	0.000	0.067	0.015	0.307
History of high blood pressure	−4.404	1.077	16.726	1	0.000	0.012	0.001	0.101
Tobacco consumption	−1.920	0.532	13.009	1	0.000	0.147	0.052	0.416
Alcohol consumption	−2.939	0.585	25.239	1	0.000	0.053	0.017	0.167
Drug use	−3.343	0.846	15.595	1	0.000	0.035	0.007	0.186
Duration of VF performing	−0.117	0.067	3.075	1	0.079	0.889	0.780	1.014
False Positive	−0.004	0.014	0.085	1	0.771	0.996	0.970	1.023
False Negative	0.031	0.011	8.905	1	0.003	1.032	1.011	1.053
Tested Points	0.000	0.002	0.016	1	0.899	1.000	0.996	1.004
Slope 10gr	0.335	0.113	8.833	1	0.003	1.397	1.121	1.742
HOV	−0.099	0.031	10.231	1	0.001	0.906	0.853	0.962
PD	0.388	0.136	8.167	1	0.004	1.475	1.130	1.925
AD	−1.103	0.399	7.634	1	0.006	0.332	0.152	0.726

**Table 9 jpm-12-00777-t009:** Comparative distribution of affirmative responses to the questionnaire, “Symptom inventory”, in the suspected OHT and NIOP groups.

Symptoms Questioned	Answers “Yes” OHT Patients (71)		Answers “Yes” NIOP Patients (749)		Answers “Yes” Total Patients (820)	
	Number	%	Number	%	Number	%
Tearing	34	28.10	15	28.31	49	28.17
Sensation of dry eyes	12	9.92	5	9.43	17	9.77
Sensation of tension in the eye-Eye strain	15	12.40	2	3.77	17	9.77
Scotomas-the lack of a part of the visual field	5	4.13	2	3.77	7	4.02
Limited view: tube/tunnel view	2	1.65	1	1.89	3	1.72
Difficulty in short distance sight	18	14.88	7	13.21	25	14.37
Difficulty in remote view (to see at a distance)	4	3.31	5	9.43	9	5.17
Disorders in color perception changes in color intensity	4	3.31	1	1.89	5	2.87
Ebluisare-blindness in bright light	14	11.56	9	16.98	23	13.22
Ebluisare-blindness when passing from light to darkness	13	10.74	6	11.32	19	10.92

**Table 10 jpm-12-00777-t010:** Distribution of binomial logistic regression parameters to the symptom questionnaire.

Symptoms Questioned	Parameter Estimate	SE	Wald ꭓ^2^	*df*	*Sig*	Exp(B)	95% CI	
							Lower	Upper
Tearing	−2.576	0.467	30.497	1	0.000	0.076	0.030	0.190
Sensation of dry eyes	−0.156	0.863	0.033	1	0.856	0.856	0.158	4.639
Sensation of tension in the eye-Eye strain	−2.379	0.956	6.201	1	0.013	0.093	0.014	0.603
Scotomas	−0.025	1.651	0.000	1	0.988	0.975	0.038	24.788
Limited view: tube/tunnel view	−0.666	2.304	0.084	1	0.773	0.514	0.006	46.942
Difficulty in short distance sight	−2.407	0.624	14.869	1	0.000	0.090	0.026	0.306
Difficulty in remote view (to see at a distance)	1.120	1.663	0.454	1	0.501	3.065	0.118	79.841
Disorders in color perception-in color intensity	−2.131	1.821	1.369	1	0.242	0.119	0.003	4.214
Ebluisare-blindness in bright light	−0.798	0.696	1.315	1	0.252	0.450	0.115	1.761
Ebluisare-blindness-passing from light to darkness	−0.736	0.798	0.851	1	0.356	0.479	0.100	2.287

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
