# Peer review of "Detection of Intraocular Hypertension during Opportunity Screening (Check-Up Medical Inspections)"

_jpm, 2022, doi:10.3390/jpm12050777_

Round 1

Reviewer 1 Report

Interesting work, which analyses a large population, with very complete statistics.
It is striking that of the 820 patients studied only 71 had ocular hypertension, it could be related to the fact that they are middle-aged patients, it would change by analysing other populations.

Author Response

Dear Reviewer

I really appreciate your recommendations. We tried, based on all the recommendations of the reviewers, to improve and correct the article.

Thank you for your support and help.

Reviewer 2 Report

Muntaenu, Muntaenu, et al. describe a retrospective epidemiological investigation of the utility of work-specific medical screenings in detecting ocular hypertension (OHT). The authors examine pneumo-tonometry as a secondary prevention tool for OHT (and potentially POAG). The authors appear to repeat (in Romania) the efforts of Yamada, Hiratsuka, et al. (2020) performed in Japan. The study and the manuscript is very intriguing, timely, and important.

My main issue with the manuscript is that it is very difficult to read. I have read it several times (and, again, the information is interesting, and I am convinced the results are truthful and reliable), but it is very difficult to follow. I HIGHLY recommend the authors work with an English-editing service to bring the manuscript up to the English standards of MDPI journals.

I do have some suggestions to improve the readability of the manuscript before the receive an extensive English edit.

Lines 1-3 (Title): I suggest something similar to that used in https://pubmed.ncbi.nlm.nih.gov/33364847/). It is confusing as written.

Lines 26-17: "TS (Transportation Safety)…" It should read "Transportation Safety (TS)" the first time.

Abstract (overall): The authors need to improve the abstract. Again, I believe the idea of the paper is to describe how the authors "captured" OHT suspects in occupational screenings, brought them to diagnosis, then they describe the parameters that best predict the diagnosis. The abstract does not describe that very well.

Lines 49-51: I suggest re-wording and removing the quotation marks. I believe the authors over-use quotation marks throughout the manuscript.

Lines 58-19: Suggest something more like "Glaucoma can be medically treated with the purpose of maintaining and restoring health."

Lines 73-81: Please edit this section. It is hard to follow as written.

Line 91: While accurate, "anamnestic" should be defined first.

Lines 98-102: This section is also very rough. Please re-write.

Table 6: HTIO? I suggest using "OHT" and being consistent throughout the manuscript.

Again, these are just some ideas. Please consult an English-editing serving for a re-write. I look forward to reading the revision. 

Author Response

Dear Reviewer

I really appreciate your recommendations. We tried, based on all the recommendations of the reviewers, to improve and correct the article.

Thank you for your support and help.

The reviwer recommendations and  our responses.

Muntaenu, Muntaenu, et al. describe a retrospective epidemiological investigation of the utility of work-specific medical screenings in detecting ocular hypertension (OHT). The authors examine pneumo-tonometry as a secondary prevention tool for OHT (and potentially POAG). The authors appear to repeat (in Romania) the efforts of Yamada, Hiratsuka, et al. (2020) performed in Japan. The study and the manuscript is very intriguing, timely, and important.

My main issue with the manuscript is that it is very difficult to read. I have read it several times (and, again, the information is interesting, and I am convinced the results are truthful and reliable), but it is very difficult to follow. I HIGHLY recommend the authors work with an English-editing service to bring the manuscript up to the English standards of MDPI journals.

I do have some suggestions to improve the readability of the manuscript before the receive an extensive English edit.

Lines 1-3 (Title): I suggest something similar to that used in https://pubmed.ncbi.nlm.nih.gov/33364847/). It is confusing as written.

Thank you for your recommendation. I made corrections.

” Detection of intraocular hypertension during oportunity screening (check-up medical inspections)

Lines 26-17: "TS (Transportation Safety)…" It should read "Transportation Safety (TS)" the first time.

Thank you for your recommendation. I made corrections.

Abstract (overall): The authors need to improve the abstract. Again, I believe the idea of the paper is to describe how the authors "captured" OHT suspects in occupational screenings, brought them to diagnosis, then they describe the parameters that best predict the diagnosis. The abstract does not describe that very well.

Thank you for your recommendation. I made corrections.

  • In Romania, the medical examination of personnel with responsibilities in transport safety is legally regulated, being mandatory as a result of the internal transposition of European legislation in the field.
  • The addressability of the patients for examination was determined by the personal choice depending on the accessibility of the medical service on the profile market (availability and price).
  • (Exp (B) = 0.782, Sig = 0.558, 95% CI = 0.343-1,783; sex is not a significant predictor at the 5% level).

Lines 49-51: I suggest re-wording and removing the quotation marks. I believe the authors over-use quotation marks throughout the manuscript.

Thank you for your recommendation. I made corrections.

Lines 58-19: Suggest something more like "Glaucoma can be medically treated with the purpose of maintaining and restoring health."

Thank you for your recommendation. I made corrections.

For OHT and GPUD, prophylaxis includes the general purposes of medicine: the promotion, maintenance and recovery of vision as far as possible [3,4].

Lines 73-81: Please edit this section. It is hard to follow as written.

Thank you for your recommendation. I made corrections.

Screening is a secondary prophylaxis action aimed at identifying the presumed persons affected by a latent health problem, so far unknown, by performing a test, an examination or other investigative techniques that can be quickly applied on masse” [5]. Early detection of diseases can be done by the following medical procedures: passive-active detection during current medical consultations, regular medical examinations at nodal ages, examination of people at risk and "Check-ups" or health check on request (for other reasons than the disease assessed in specialized services at the initiative of the beneficiary [5], or as a legal obligation [6,7]. “Health check on request screening is as effective as mass screening, but it is limited to the population that attends health services” [8,9].

Line 91: While accurate, "anamnestic" should be defined first.

Thank you for your recommendation. I made corrections.

We replaced ”anamnestic” with ”personal medical history”.

Lines 98-102: This section is also very rough. Please re-write.

Thank you for your recommendation. I made corrections.

The screening had a legally required periodicity of 1 year and consisted of examinations in 7 medical specialties: Internal Medicine, Surgery, Otorhinolaryngology, Ophthalmology, Neurology, Psychiatry and Laboratory Medicine [6,7].

Table 6: HTIO? I suggest using "OHT" and being consistent throughout the manuscript.

Thank you for your recommendation. I made corrections.

Again, these are just some ideas. Please consult an English-editing serving for a re-write. I look forward to reading the revision. 

Thank you for your recommendation and help.

Reviewer 3 Report

The authors performed a oportunity screening to for OHT early detection, and further investigated the determinants of OHT. The beauty of the study was the comprehansive medical and opthalmic examiniations, for detailed data collection. However, I think the major limitation of the study is lack of novelty to the literature, as studies for glaucoma screening have been well report, and the advantage of the current screening strategy has not been well described. And I think the English writing must be imporved. Sometime I cannot understand what the authors try to say. 

Other comments: 

  1. Line 31: could the sex ratio result in bias?
  2. Line: Please explain ''Check-ups" and ''Opportunity screening'' in line 67.
  3. Line 105: OHT syndrome determined by only one measurement may not be a rigorous design, as IOP may fluctuate diurnally. 
  4. Line 138-140: I don't think the it is a standard method to assess the patients visual field. And How to define a patient with or without glaucomatous visual field defect? 
  5. Line 145: Please check the manuscript and spell out the abbreviations which come up at the first time. 
  6. Discussion: I think the author shall additionally discuss the potential reasons of why the results showed significant associations (e.g. difficulty in remote view to OHT), rather than emphasizing the importance of early detection of OHT and glaucoma which has been well recognized. 

Author Response

Dear Reviewer

I really appreciate your recommendations. We tried, based on all the recommendations of the reviewers, to improve and correct the article.

Thank you for your support and help.

The reviwer recommendations and  our responses.

The authors performed a oportunity screening to for OHT early detection, and further investigated the determinants of OHT. The beauty of the study was the comprehansive medical and opthalmic examiniations, for detailed data collection. However, I think the major limitation of the study is lack of novelty to the literature, as studies for glaucoma screening have been well report, and the advantage of the current screening strategy has not been well described. And I think the English writing must be imporved. Sometime I cannot understand what the authors try to say. 

The advantage of the current screening strategy lies in 3 essential elements: the obligation to perform regular examinations, funding by the beneficiary and the complex and unitary examination protocol for all medical units (in Romania in 2014-2022 were approved by the Ministry of Transport and Infrastructure a number of 1210 medical and psychological units.[43].

The specification of the diagnosis of OHT / POAG is no longer limited by mandatory procedures.

The novelty that this type of screening could bring in the medical literature could be given by the opportunity of a specialized study at national level that would benefit from four great advantages: unitary methodology, obligatory application at national level, comparability of statistical data , the unitary evaluation of the health condition of the personnel in this field of activity.

The state intervention was exercised only at two levels: the issuance of the normative act, the approval activity of the medical units and the supervision of their activity through the annual approval procedure.

Other comments: 

1. Line 31: could the sex ratio result in bias?

Thank you for your recommendations. I made corrections.

The study involved 754 men (92.0%) and 66 women (8.0%), the sex ratio is 11.42 (Exp (B) = 0.782, Sig = 0.558, 95% CI = 0.343-1,783; sex is not a significant predictor at the 5% level).

2. Line: Please explain ''Check-ups" and ''Opportunity screening'' in line 67.

Thank you for your recommendations. I made corrections.

“Health check on request screening is as effective as mass screening, but it is limited to the population that attends health services” [8,9].

3. Line 105: OHT syndrome determined by only one measurement may not be a rigorous design, as IOP may fluctuate diurnally. 

Thank you for your recommendations. I made corrections.

The IOP determination was performed at two times: in the morning between 8-10 and then between 13-14. Each measurement expressed the average of three trials for each eye. The most favorable result was chosen.

4. Line 138-140: I don't think the it is a standard method to assess the patients visual field. And How to define a patient with or without glaucomatous visual field defect? 

"Optopol PTS 910 - Automated Perimeter" calculates statistically and graphically shows the centralized defect of the test result "The Bebie Curve" (indicating VF quality) which is constructed by ordering the sensitivities of each tested point in descending order. According to the instruction manual, "Bebie Curve" is particularly useful in quickly and immediately classifying the type of defect, especially in differentiating a diffuse loss from a localized or mixed one. [16].

For the statistical interpretation of the graph of the centralized defect of a test result (“Bebie Curve”) we used the following categorical classification (Table 1) [15].

Thank you for your recommendations. I made corrections.

5. Line 145: Please check the manuscript and spell out the abbreviations which come up at the first time. 

Thank you for your recommendations. I made corrections.

Check-up medical examinations the “Medical Opinion” was issued, which together with the “Psychological Opinion” documented the issuance of the work aptitude certificate by the Labor Medicine specialist [17].

6. Discussion: I think the author shall additionally discuss the potential reasons of why the results showed significant associations (e.g. difficulty in remote view to OHT), rather than emphasizing the importance of early detection of OHT and glaucoma which has been well recognized. 

Thank you for your recommendations. I made corrections.

The average age of the studied groups was 43.27± 9.92, which is the age at which refractive issues necessitate adequate optical correction. We point out that the minimum essential distance visual acuity standards, both with and without optical correction, are stringent in order to gain ophthalmology approval [12].

Although not all patients mentioned in the questionnaire the aspect related to the difficulties encountered in distance vision, the statistical analysis of the VA test results shows that 235 patients (28.5%) of those investigated need optical correction, of which 25 patients (10.6%) with OHT compared to 210 patients (89.4%) without OHT.

Completion in the subsection of statistical analysis

…the binary logistic regression is used to develop a model by which the values of the criterion (dependent variable) are probabilistically associated with the values of the predictors (independent variables) to identify probabilities.

Also, the analysis of the regression model of VF parameters specifies a probabilistic association of OHT with predictor values at a statistically significant level: PD: Exp.(B)=1.475, Sig=0.004, 95% CI=1.130-1.925; Slope at 100: Exp.(B)=1.397, Sig=0.003, 95% CI=1.121-1.742; False negative: Exp.(B) =1.032, Sig=0.003, 95% CI =1.011-1.053 and AD Exp.(B) = 0.332, Sig=0.006, 95% CI =0.152-0.726.

Round 2

Reviewer 2 Report

The manuscript is much-improved. Please make the following changes before final consideration:

  1. Line 3: The correct spelling is "opportunity" (not "opportunity").
  2.  Lines 316-319: "..., which is the age at which..." is confusing. I assume the authors mean to say, "All the symptoms investigated were present in a higher absolute value in the group of OHT suspects, except for “Difficult to see at a distance.” The average age of the participants was 43.27± 9.92, which is consistent with the age at which refractive issues necessitate some form of optical correction."
  3.  All tables: I highly recommend that the headings (titles) for all tables are left justified on top of the tables. For example, for Table 10...

Table 10. Distribution of ...

___________________________________________________________________________

___________________________________________________________________________

I also HIGHLY suggest that a very thorough grammar check is done to eliminate any missing punctuation, etc.

Final note. This is a very interesting paper. Best wishes for success on its distribution, citing, etc. It is very intriguing and worthwhile.

Author Response

(The authors gave the same response as above.)

Reviewer 3 Report

I have no further comment.

Author Response

(The authors gave the same response as above.)
